# Fine-tuning a Foundation Model for Fetal Orientation Prediction From Blind-Sweep Ultrasound Videos

**Chun Kit Wong**[*1,2]                                        CKWO@DTU.DK
**Mary Le Ngo**[*3,4,5]                        MARY.VAN.ANH.LE.NGO.01@REGIONH.DK
**Jakob Ambsdorf**[2,3]                                        JAAM@DI.KU.DK
**Anders Nymark Christensen**[1]                               ANYM@DTU.DK
**Mads Nielsen**[2,3]                                          MADSN@DI.KU.DK
**Martin Grønnebæk Tolsgaard**[3,5]    MARTIN.GROENNEBAEK.TOLSGAARD@REGIONH.DK
**Aasa Feragen**[1,2]                                          AFHAR@DTU.DK

[1] *Technical University of Denmark, Kongens Lyngby, Denmark*

[2] *Pioneer Centre for AI, Copenhagen, Denmark*

[3] *University of Copenhagen, Copenhagen, Denmark*

[4] *Slagelse Hospital, Slagelse, Denmark*

[5] *CAMES Rigshospitalet, Copenhagen, Denmark*

**Editors:** Accepted as short paper at MIDL 2026

## Abstract

Previous work on determining fetal orientation from blind ultrasound sweeps has relied on anatomical heuristics, specifically fetal head detection and segmentation. However, clinical validation reveals that this approach is not robust, frequently failing when the head is malpositioned. To address this, we propose a temporally-aware framework that leverages a foundation model to encode video frames into a 768-dimensional embedding space, followed by an ensemble of 1D-convolutional classification heads to predict fetal orientation directly. Code available at https://github.com/wong-ck/ultradino-navigator-fetal-us.

**Keywords:** fetal ultrasound, navigation, foundation model

## 1. Introduction

To support clinicians in navigating to good ultrasound standard plane, Wiśniewski et al. (2025) established a heuristic-based approach for determining fetal orientation from blind ultrasound sweeps. The method works by first localizing the fetal head using a segmentation model, then applying heuristics to determine presentation and lie. If the head appears earlier during the caudal-cranial sweep, the presentation is classified as cephalic; otherwise, it is classified as breech. For lie classification, the method relies on the fact that the thalamus is situated posteriorly to the cavum septum pellucidum within the fetal head.

However, subsequent clinical validation on a broader patient cohort has revealed a critical failure mode: a lack of robustness stemming from the system's strict dependency on successful fetal head detection. Although the method is accurate when the head is localized, inability to detect the head results in a refusal to provide a prediction. This behavior mirrors the challenges observed in the deployment of diabetic retinopathy screening tools,

---

* Contributed equally

where abstaining from making predictions on *unreadable* images significantly hindered clinical workflow and utility (Beede et al., 2020).

The failure to detect the fetal head is primarily driven by two factors: anatomical variability and gestational progression. In breech presentations, where the fetus is positioned upright, the head often lies outside the maternal midline, eluding the field of view of a standard sweep. Furthermore, in late-trimester cephalic presentations, the fetal head frequently descends into the pelvic cavity in preparation for delivery, obscuring the standard anatomical planes required for heuristic-based identification.

## 2. Proposed Method

To overcome these limitations, we transition from anatomy-specific heuristics to a robust, temporally-aware deep learning framework that predicts fetal orientation directly from sweep sequences. This allows the network to leverage alternative landmarks, such as the fetal abdomen or spinal column, to infer orientation when the head is not visible. Given the high data requirements of full 3D architectures, we adopt a more efficient spatio-temporal approach. We use a pre-trained 2D encoder to project each video frame into a compact embedding space, followed by a sequence classification head that operates on the resulting 2D embedding-temporal dimensions. Our implementation leverages UltraDino (Ambsdorf et al., 2025), a ViT-B/16-based foundation model pre-trained on fetal ultrasound images, to encode each frame of a sweep video into a 768-dimensional embedding. For a sequence of 100 frames subsampled from a 10-second sweep, the resulting $768 \times 100$ feature matrix is processed by multiple classification heads, each comprising a 1D-convolutional layer. These heads are trained to predict fetal presentation and lie separately. To ensure generalizability, we employ a 4-fold cross-validation strategy, training four separate models that are subsequently integrated into an ensemble for inference.

### 2.1. Abstention for Ambiguous Sweeps

We also address the challenge of poor-quality sweeps where even clinical experts cannot determine fetal orientation. To handle this ambiguity, we adopt an uncertainty quantification approach by labeling these samples as *unknown*. During training, instead of treating *unknown* as a discrete class, we maximize the entropy of the softmax logits for these samples. Post-training, we determine an optimal entropy threshold that maximizes the F1-score by allowing the model to abstain from uncertain samples. This approach is motivated by the intuition that, given our limited dataset size, an explicit *background* class would be insufficient to capture the vast and varied appearance of uninterpretable scans.

## 3. Experimental Dataset

This study utilizes a private dataset of 188 sweep videos from 44 unique pregnant women, acquired at Rigshospitalet, Denmark (the SONAI dataset). Each sweep was manually annotated for fetal presentation and lie. Expanding on the binary lie classifications (left/right) of Wiśniewski et al. (2025), we introduce *anterior* and *posterior* classes to provide a more granular description of fetal orientation. To account for postural nuances, such as head rotation relative to the torso, we recorded two separate lie annotations per sweep: one for

the head and one for the abdomen. We also sampled and annotated 91 midline sweeps from the public ACOUSLIC dataset (Sappia et al., 2025) using the same protocol. An 80 : 20 patient-level train-test split was performed independently for each dataset before merging.

## 4. Results and Discussion

| | | Presentation | | | | | | | Lie (Head) | | | | | | | | | | Lie (Abdomen) | | | | |
| | | Heuristic | | | Ultradino | | | | Heuristic | | | | | Ultradino | | | | | Ultradino | | | | |
| | True (↓) \ Pred (→) | unk | cep | bre | unk | cep | bre | True (↓) \ Pred (→) | unk | pos | ant | lef | rig | unk | pos | ant | lef | rig | unk | pos | ant | lef | rig |
|---|---|---|---|---|---|---|---|---|---|---|---|---|---|---|---|---|---|---|---|---|---|---|---|
| SONAI | | | | | | | | unk | 10 | 0 | 0 | 0 | 0 | 3 | 0 | 0 | 0 | 7 | 2 | 0 | 0 | 3 | 2 |
| | unk | 1 | 0 | 0 | 0 | 1 | 0 | pos | 3 | 0 | 0 | 0 | 0 | 1 | 0 | 0 | 0 | 2 | 0 | 1 | 0 | 1 | 2 |
| | cep | 19 | 10 | 0 | 0 | 29 | 0 | ant | 5 | 0 | 0 | 0 | 0 | 1 | 0 | 0 | 0 | 4 | 1 | 0 | 0 | 1 | 0 |
| | bre | 10 | 0 | 1 | 1 | 0 | 10 | lef | 3 | 0 | 0 | 2 | 1 | 2 | 0 | 0 | 4 | 0 | 2 | 0 | 0 | 10 | 0 |
| | | | | | | | | rig | 14 | 1 | 0 | 0 | 2 | 2 | 0 | 0 | 0 | 15 | 1 | 0 | 0 | 2 | 13 |
| ACOUSLIC | | | | | | | | unk | 10 | 0 | 1 | 0 | 0 | 8 | 0 | – | 0 | 3 | 0 | 0 | – | 1 | 2 |
| | unk | 4 | 0 | 0 | 1 | 3 | 0 | pos | 2 | 0 | 0 | 0 | 0 | 1 | 0 | – | 0 | 1 | 3 | 0 | – | 0 | 1 |
| | cep | 7 | 5 | 0 | 1 | 11 | 0 | ant | 0 | 0 | 0 | 0 | 0 | – | – | – | – | – | – | – | – | – | – |
| | bre | 3 | 0 | 0 | 3 | 0 | 0 | lef | 3 | 0 | 0 | 2 | 0 | 3 | 0 | – | 2 | 0 | 5 | 0 | – | 2 | 0 |
| | | | | | | | | rig | 1 | 0 | 0 | 0 | 0 | 1 | 0 | – | 0 | 0 | 2 | 0 | – | 1 | 2 |

Table 1: Confusion matrices showing method comparison [Heuristic, Ultradino] grouped in column by task [Presentation, Lie (Head), Lie (Abdomen)] and in row by dataset [SONAI, ACOUSLIC]. Lie (Abdomen) shows Ultradino results only, since it is not supported by the heuristic method. Abbreviations: unk=unknown, cep=cephalic, bre=breech, pos=posterior, ant=anterior, lef=left, rig=right.

The classification performance of our ensemble model is summarized in Table 1. Compared to the heuristics-based columns, the Ultradino-based columns demonstrate high fidelity on the SONAI dataset, particularly for fetal presentation. In contrast, performance on the ACOUSLIC dataset was notably lower. We attribute this discrepancy to the inherently lower image quality and different acquisition conditions of the ACOUSLIC sweeps, which likely hindered the UltraDino encoder's ability to extract clear anatomical embeddings. For instance, in the ACOUSLIC presentation task, a higher proportion of breech samples were classified as *unknown*, reflecting the model's increased uncertainty when faced with degraded visual features.

Regarding fetal lie prediction, the model effectively distinguishes between *left* and *right* orientations, especially in the abdomen-based task. However, performance on the *posterior* and *anterior* classes remains poor. As shown in the *Lie Head* and *Lie Abdomen* matrices, the model frequently misclassifies these cases, often labeling them as *unknown* or defaulting to the more prevalent left/right classes. This is largely due to severe class imbalance in our dataset; the limited number of anterior and posterior samples prevented the model from learning the specific spatio-temporal features associated with these orientations.

## Acknowledgments

This work is funded by the Danish Pioneer Centre for AI (DNRF grant number P1), SONAI - a Danish Regions' AI Signature Project, and the Novo Nordisk Foundation through the Center for Basic Machine Learning Research in Life Science (NNF20OC0062606).

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
