# OpenReview forum: "Fine-tuning a Foundation Model for Fetal Orientation Prediction From Blind-Sweep Ultrasound Videos"
_MIDL.io/2026/Short_Papers — MIDL 2026 - Short Papers Poster_

### Official Review · Reviewer_bZqt · 2026-05-04
**Simple yet interesting paper**

**Rating:** 4
**Confidence:** 4

**Review:**

see below.

**Summary:**

This paper introduces a deep learning framework to replace previous existing methods for fetal head and abdomen orientation classification from blind ultrasound sweeps.
The model relies on a foundation model that is complemented with an ensemble of classifications heads. This simple method is shown to outperform previous heuristics-based methods in-domain, but has more problems generalising to out-of-domain data.

**Strengths:**

1. interesting paper that is addressing an overlooked problem by the community.

2. the method is simple but easy to follow, despite some missing details (see below)

3. the method yields promising results, especially in-domain, where it outperforms previous approaches.

4. well-motivated paper.

**Weaknesses:**

1. some details are missing for the method, for example the training loss and some basic pre-processing/augmentation information, as well as for the dataset (MRI sequence, resolution, etc.).

2. Out-of-domain results are poor, which I guess is expected since there is no real strategy to handle generalisation issues. Nonetheless, this should be listed as a limitation.

3. The authors frame this problem as classification, but it seems pretty coarse to me (right/left, anterior/posterior, cephalic/breech). They should probably compare/acknowledge/discuss existing methods in pose estimation.

4. Some figures are clearly missing to illustrate the strategy and some outcomes.

**Justification Of Rating:**

I think this work will be a good addition for the MIDL conference, but I encourage the authors to address some of my points in the final version.

---

### Decision · Program_Chairs · 2026-05-08

Accept (Poster)